# Transfer Learning Across Datasets with Different Input Dimensions: An Algorithm and Analysis for the Linear Regression Case

## Abstract

With the development of new sensors and monitoring devices, more sources of data become available to be used as inputs for machine learning models. These can on the one hand help to improve the accuracy of a model. On the other hand, combining these new inputs with historical data remains a challenge that has not yet been studied in enough detail. In this work, we propose a transfer learning algorithm that combines the new and historical data with different input dimensions, which is especially beneficial when the new data is scarce. We focus the approach on the linear regression case, which allows us to conduct a rigorous theoretical study on the benefits of the approach. Our approach achieves state-of-the-art performance on several real-life datasets, outperforming other linear transfer learning algorithms and performing comparably to non-linear ones. In addition, we prove that our approach is robust against negative transfer learning assuming that the new inputs are normally distributed, and confirm its robustness empirically also on real-world data distributions.

## 1 Introduction

The constant evolution of sensor technology and measuring equipment brings ever more data sources that can be employed by machine learning practitioners to build better predictive models. In the healthcare domain, for example, new ICT equipment is installed and starts generating new sensory data that helps doctors to make better diagnostics (Sheng & Ling, 2006). Similarly, in the predictive maintenance domain, new sensors are developed and installed to help monitoring the state of industrial equipment. In both cases, it is desirable to update the predictive model or to train a new one so that it can make use of the new inputs. However, the data collection can be expensive and time-consuming, taking a long time before it is feasible to train a new model. This can be seen as a transfer learning problem where there are two datasets: the historical data with plenty of samples but without the new input features and the newly collected data with fewer samples, but with all available input features (Figure 1). The goal then is to use the historical data as the source dataset to improve the prediction accuracy using all inputs, which are observed only in the target dataset. We refer to this setup as "incremental input transfer learning".

Research in transfer learning and domain adaptation has put forward methods for learning a target task with limited data available by using data from other similar tasks referred to as source tasks. For example, recent transfer learning works (WEI et al., 2018) yield state-of-the-art accuracy on classifying images from a target domain given only a small portion of target data. These works have considered mainly two variations of this setting: transfer across different tasks and transfer across different input domains (Pan & Yang, 2010). For the latter case, most research works assume that the inputs have the same dimension and come from different distributions, exploiting some semantic similarity between the domains (Chen et al., 2021; 2015; Obst et al., 2021; Mousavi Kalan et al., 2020; Zhuang et al., 2021). Although these works assume arbitrary distributions for source and target domains, it is non-trivial to show that their results generalize when the input dimensions are different. Other works use mapping functions (Yang et al., 2016; Moon & Carbonell, 2017; He et al., 2020; Yan et al., 2018; Wei et al., 2019) to cast the data into a new domain where the

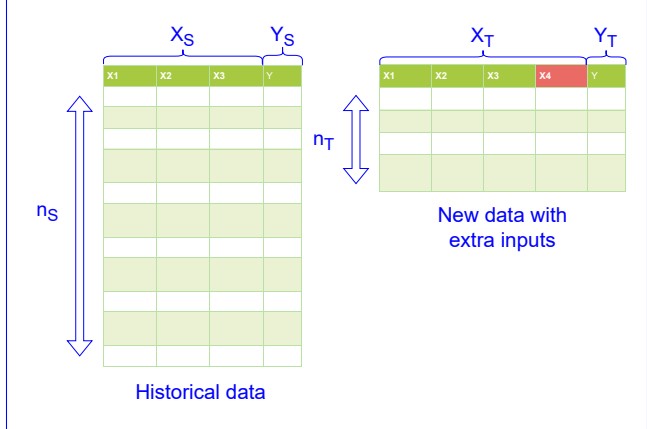

Figure 1: The historical and the new datasets represented as a transfer learning problem where the target differs by the amount of inputs.

source and target data are compatible, but we argue that for this kind of approach to work it requires some exploitable relationship between historical and new features, whereas our approach also works when there is no such relationship or when it is weak.

In this paper, we study the incremental input problem theoretically in its linear regression version. We summarize our contributions in the following items:

- We provide an efficient and easy to implement transfer learning approach for this problem which is especially helpful when the new input data is scarce;

- We show through rigorous theoretical study that the approach is robust against negative transfer learning[1] and we prove an upper bound for its generalization error;

- We confirm empirically the robustness of our approach using real-life data and show that it outperforms other linear mapping based transfer learning approaches, and performs on par with more complex state-of-the-art non-linear approaches.

## 2 Related Work

**Transfer Learning:** Studies in transfer learning seek to learn a target task with limited data by using large amounts of data from source datasets with similar labels or input features (Pan & Yang, 2010). In the linear regression case, Obst et al. (2021) propose the notion of transfer gain and a statistical test to tell when using a source dataset to pre-train a model is better than training a model from scratch. They assume that the model should be fine-tuned by using gradient descent on the target data, but often this approach is less efficient than the least-squares method since it requires carefully selecting the step size and number of iterations, and can lead to sub-optimal solutions. Chen et al. (2015) present a linear regression method that combines two datasets: one is small and unbiased, and the other is large and biased. They study in detail the conditions when their approach is beneficial as a function of the amount of bias in the large dataset. Other works focus on the theoretical properties of transfer learning: Mousavi Kalan et al. (2020) show the minimax bounds for transfer learning based on a notion of true distributions where the source and target datasets are sampled from; and Hanneke & Kpotufe (2019) propose a new discrepancy metric between source and target data and prove the minimax bounds for learning a classifier according to their metric. Their setting differs from ours mainly because we assume that one or more variables are not observed in the large dataset, while they assume that all variables are present in both datasets.

---

[1]According to the transfer gain definition from Obst et al. (2021).

**Heterogeneous Transfer Learning:** This branch of transfer learning concerns the problem where the inputs in the source and the target domains differ in feature dimensionality (Zhuang et al., 2021). Here, the main focus of existing approaches is on learning a feature mapping from the source and target datasets into a homogeneous feature space where all data can be combined to train the final model. They split in supervised (Yan et al., 2018; Moon & Carbonell, 2017) and unsupervised (Yang et al., 2016; He et al., 2020; Wei et al., 2019).

In the supervised case, Yan et al. (2018) uses class labels to improve the distribution alignment of the source and target data in the new feature space. On top of learning the feature mapping, Moon & Carbonell (2017) also uses an attention mechanism to weight the source instances based based on their corresponding classification accuracy in a joint label space. We do not compare to these methods since they require classification labels, and we focus on regression tasks.

In the unsupervised case, some heterogeneous transfer learning approaches rely on instance-correspondence (IC) between pairs of source and target data to learn the feature mapping (Yang et al., 2016; He et al., 2020). For example, for text sentiment classification tasks where source and target data are in different languages, a document in the target language corresponds to its translation in the source language (Yang et al., 2016). However, this kind of data pairs do not exist in many domains (i.e. medical records or predictive maintenance data), rendering IC approaches unfeasible in those cases.

Another unsupervised approach by Wei et al. (2019) assumes a common subset of features among source and target data and uses it to estimate the value of the missing target-specific features through a feature mapping function. The mapping is obtained by minimizing the following objective:

$$\min_W \|f(x_T^{\mathrm{hist}}, W) - x_T^{\mathrm{new}}\|^2 + \alpha \mathrm{MMD}(f(x_S, W), x_T^{\mathrm{new}}) + \beta\|W\|^2$$

where $x_T^{\mathrm{hist}}$ and $x_T^{\mathrm{new}}$ are the historical features and the new features, respectively, which are observed in the target dataset, while $x_S$ is the source dataset where only the historical features are observed and $W$ are the DSFT parameters. The first term corresponds to a least-squares regression from the historical features to the new ones, the second term is the maximum mean discrepancy (MMD) between the features predicted for the source dataset and the ones observed in the target dataset. The last term is a simple L2 regularization of the parameters. They propose two versions to apply their method: one where $f$ is just a linear mapping ($\mathrm{DSFT}_l$) and a kernelized variation of it ($\mathrm{DSFT}_{nl}$), and they present closed-form solutions for each version. In the incremental input learning case, our final goal is to learn a predictor of the labels using all the target features. This approach works as a preprocessing step to fill up the source dataset, but if the mapping between historical and new features is not significantly represented in the data or too difficult to learn (i.e. when the target data is limited and/or the source data is very large), DSFT can introduce extra noise and hurt the performance of the predictor. Nevertheless, we compare DSFT with our approach in our experiments in the later sections.

**Incremental attribute learning:** The problem of learning a model when the number of inputs is changing has been studied previously by Guan & Li (2001). They propose an algorithm to automatically update a neural network when new inputs are discovered. However, as their work's main focus is to propose a new algorithm and evaluate it empirically, there is no study about its theoretical properties. In addition, they assume sufficient data to train a neural network from scratch using the new features, which may, in many cases, mean that the historical data is no longer necessary.

Also similar to incremental learning, Hou et al. (2017); Zhang et al. (2020) study the problem of online learning where the data comes in streams and features are added and removed over time. Their setting assumes that only one data point is observed at each time and while the previously streamed data are lost, making their approach significantly less applicable to our transfer learning problem.

**Missing data:** Another way of approaching the incremental learning problem is by treating the new features as missing in the source dataset and solving it by using missing data techniques, widely documented by Little & Rubin (2019). The most commonly used approaches in this domain, however, are designed for when the proportion of missing data is small, while in this paper we focus on the case where the vast majority of data points have the same missing features.

## 3 Defining a linear model for the source and the target datasets

We are concerned with the problem of learning a model based on two datasets: the historical data and the newly collected data containing extra features. We will refer to the first one as the source dataset and to the second one as the target dataset since we want to tackle this problem using a transfer learning approach. We choose linear regression for this study because it allows us to derive closed-form solutions that are easier to inspect and gain insights into the nature of the problem. The labels in both datasets represent the same linear regression task and follow the same prior distribution. In this section, we want to give a formal definition of these datasets and their conditional distributions assuming the context of linear regression. These definitions will be used throughout the remainder of the paper.

We define the target dataset as $(x_T, \boldsymbol{Y_T})$, where $x_T \in \mathbb{R}^{n_T \times d_T}$ is a full rank design matrix containing $n_T$ independent observations of $d_T$ input features and $\boldsymbol{Y_T} \in \mathbb{R}^{n_T}$ is a random vector containing the respective labels. They are related by the linear model $\boldsymbol{Y_T} = x_T \theta + \boldsymbol{w_T}$, where $\theta$ is the $d_T$-dimensional parameter vector describing the linear relationship between inputs and labels, and $\boldsymbol{w_T} \sim \mathcal{N}(\boldsymbol{0}_{n_T}, \sigma^2 I_{n_T})$ is additive Gaussian noise. Our goal is to learn the parameter vector $\theta$. In addition, we define the source dataset as $(x_S, \boldsymbol{Y_S})$, where $x_S \in \mathbb{R}^{n_S \times d_S}$ is a full rank matrix containing $n_S$ observations of $d_S$ input features ($d_S < d_T$) and $\boldsymbol{Y_S} \in \mathbb{R}^{n_S}$ represents the random vector of the labels. For this dataset, we assume that the relationship between the labels and inputs is $\boldsymbol{Y_S} = x_S \theta' + \boldsymbol{X''}\theta'' + \boldsymbol{w_S}$, where $\theta'$ and $\theta''$ correspond respectively to the first $d_S$ and last $d_T - d_S$ components of $\theta$ and $\boldsymbol{X''}$ is a $n_S \times (d_T - d_S)$ random matrix with independent entries such that $\boldsymbol{X''_{ij}} \sim \mathcal{N}(0, 1)$. Again we assume additive Gaussian noise $w_S \sim N(0, \sigma^2 I_{n_S})$, independent of $w_T$. We choose these assumptions to emulate the idea that in the source dataset we can observe only part of the features available in the target dataset ($d_S$ out of $d_T$), and these features should have the same influence on the label for both datasets, represented by $\theta'$. The unobserved features are then replaced by random values so their influence on the label can also be taken into account by the model. In the later sections, we show empirically that the assumption about $\boldsymbol{X''}$ can be relaxed to other distributions. By simple probability manipulations we can derive the distribution of the source labels $\boldsymbol{Y_S}$ as:

$$\boldsymbol{Y_S} \sim \mathcal{N}(x_S \theta', (\sigma^2 + \|\theta''\|^2)I_{n_S}) \tag{1}$$

A first insight that we can gain from this formalization is that, for the source dataset, the unobserved features $\boldsymbol{X''}$ add up with the noise $\boldsymbol{w_S}$ of the source label, increasing its variance by $\|\theta''\|^2$. Based on this insight, we will refer to the noise variance of the source labels as $\sigma_S^2 = \sigma^2 + \|\theta''\|^2$.

**Basic estimator:** As we want to measure the performance improvement from using the source dataset, we select a model which uses only the target data as a baseline. We select the ordinary least-squares (OLS) estimator, which is computed by minimizing the residual sum of squares of the target data: $\mathcal{R}_T(\theta) = \|\boldsymbol{Y_T} - x_T\theta\|^2$; therefore, it is defined as $\hat{\theta}_T = (x_T^\top x_T)^{-1} x_T^\top \boldsymbol{Y_T}$. We will also refer to it as the *basic estimator*. We choose the OLS as a baseline because it is the maximum likelihood estimator of $\theta$ using $(x_T, \boldsymbol{Y_T})$ and is widely used to solve linear regression problems. Two known characteristics of the OLS estimator are that it is unbiased: $\mathbb{E}[\hat{\theta}_T] = \theta$; and its variance has a simple analytical form: $\mathrm{Var}(\hat{\theta}_T) = \sigma^2 (x_T^\top x_T)^{-1}$. These are important because they permit us to make a variance comparison with the transfer learning approach that we introduce in the following section.

At this point, we have formally defined the source and target datasets and their distributions, as well as a baseline model. With these definitions, we can now introduce a transfer learning approach for the incremental input problem.

## 4 Data-pooling estimator

Based on the previously defined source and target datasets, we want to define a transfer learning approach that combines both to produce a better model than the basic estimator. We do so by defining a loss that is a function of both datasets and deriving its solution which leads to our data-pooling estimator. We also study some properties of this solution and link it to the maximum likelihood approach, which leads to an efficient way of estimating the necessary hyperparameters. In the end, we put all these findings together into a transfer learning algorithm for the incremental input setting.

## 4.1 The data-pooling loss

A common transfer learning approach is to learn by minimizing the convex sum of errors in both datasets (Ben-David et al., 2010). In the linear regression case, this approach has been referred to as data-pooling (Chen et al., 2015; Obst et al., 2021). We define our data-pooling loss $\mathcal{R}_\alpha(\theta)$ as the weighted sum of the losses over the source and the target datasets (respectively $\mathcal{R}_S$ and $\mathcal{R}_T$), where $\alpha = (\alpha_S, \alpha_T)$ is the vector of positive weights:

$$\mathcal{R}_\alpha(\theta) = \alpha_S \mathcal{R}_S(\theta) + \alpha_T \mathcal{R}_T(\theta) \tag{2}$$

Intuitively, when $\alpha_T \gg \alpha_S$, the error in the target dataset is scaled by a factor larger than the error in the source dataset and the solution obtained by optimizing $\mathcal{R}_\alpha$ will be closer to the basic estimator. On the other hand, when $\alpha_T \ll \alpha_S$, then $\mathcal{R}_S$ will dominate the loss, and the optimal solution will distantiate from the basic estimator.

In the incremental input case, we need to make sure that the parameters related to the new inputs will not influence the error calculated on the source dataset. To achieve that, we define $\mathcal{R}_S(\theta) = \|\boldsymbol{Y_S} - x_S \mathbb{I}^\top \theta\|^2$, where $\mathbb{I}$ describes a $d_T \times d_S$ matrix such that $\mathbb{I}_{ii} = 1$ and $\mathbb{I}_{ij} = 0$ when $i \neq j$. In practice, $x_S \mathbb{I}^\top$ can be interpreted as filling up the missing dimensions of $x_S$ with zeros. By replacing the definitions of $\mathcal{R}_S$ and $\mathcal{R}_T$ in Equation 2 we obtain:

$$\mathcal{R}_\alpha(\theta) = \alpha_S \|\boldsymbol{Y_S} - x_S \mathbb{I}^\top \theta\|^2 + \alpha_T \|\boldsymbol{Y_T} - x_T \theta\|^2 \tag{3}$$

**Proposition 4.1** *Given a source and a target dataset as defined in Section 3, the data-pooling loss $\mathcal{R}_\alpha(\theta)$ is convex for any choice of $\alpha_S, \alpha_T \in \mathbb{R}^+$. Therefore it has a unique minimizing solution which is defined by:*

$$\hat{\theta}_\alpha = (\alpha_S \mathbb{I} x_S^\top x_S \mathbb{I}^\top + \alpha_T x_T^\top x_T)^{-1} (\alpha_S \mathbb{I} x_S^\top \boldsymbol{Y_S} + \alpha_T x_T^\top \boldsymbol{Y_T}) \tag{4}$$

The result of proposition 4.1 (proved in the supplemental material) gives a direct form of computing the data-pooling estimator $\hat{\theta}_\alpha$ and an analytical formula which can also be used for further analysis of the method. Based on it, and what is known about the distributions of $\boldsymbol{Y_T}$ and $\boldsymbol{Y_S}$, we are able to derive closed forms of the expected value and the variance of $\hat{\theta}_\alpha$.

**Proposition 4.2** *The data-pooling estimator is an unbiased estimator of $\theta$ (proof in the supplemental material). Its expectation and variance are:*

$$\mathbb{E}[\hat{\theta}_\alpha] = \theta \tag{5}$$

$$Var(\hat{\theta}_\alpha) = M^{-1}(\alpha_S^2 \sigma_S^2 \mathbb{I} \Sigma_S \mathbb{I}^\top + \alpha_T^2 \sigma^2 \Sigma_T) M^{-1} \tag{6}$$

*where $\Sigma_S = x_S^\top x_S$, $\Sigma_T = x_T^\top x_T$ and $M = \alpha_S \mathbb{I} \Sigma_S \mathbb{I}^\top + \alpha_T \Sigma_T$.*

The fact that $\hat{\theta}_\alpha$ is unbiased guarantees that it converges to the real parameters $\theta$, regardless of the choice of $\alpha$. Its variance, however, is influenced by $\alpha$, so we are interested in selecting the hyperparameter $\alpha$ in a way that the variance is minimal. The relationship between $\alpha$ and $Var(\hat{\theta}_\alpha)$ is complex, as Equation 6 shows, so choosing it is not trivial.

## 4.2 The relationship of data-pooling and maximum likelihood

A natural way of estimating $\theta$ is by maximizing the likelihood of the source and target labels and observations given that we know their probability distributions and also the distribution of the unobserved features $\boldsymbol{X''}$. By looking at the negative log-likelihood function of the labels $\boldsymbol{Y_S}$ and $\boldsymbol{Y_T}$ given the observations $x_T$ and $x_S$, we arrive at the following equation:

$$\mathcal{L} = \frac{1}{2\sigma_S^2} \|\boldsymbol{Y_S} - x_S \theta'\|^2 + \frac{1}{2\sigma^2} \|\boldsymbol{Y_T} - x_T \theta\|^2 + \frac{n_S}{2} \log(\sigma_S^2) + \frac{n_S + n_T}{2} \log(2\pi) + \frac{n_T}{2} \log(\sigma^2) \tag{7}$$

Finding the maximum likelihood estimator (MLE) of $\theta$ by minimizing the equation above is a complex non-linear problem. We observe that if we take the data-pooling loss $\mathcal{R}_\alpha$ with $\alpha_T = \frac{1}{\sigma^2}$ and $\alpha_S = \frac{1}{\sigma_S^2}$, then we

---

**Algorithm 1** Data-pooling estimator

---

**Input:** source dataset $(x_S, y_S)$, target dataset $(x_T, y_T)$
**Initialization:** $\bar{x}_T = \mathbf{0}_{d_T}$
$\bar{x}_{jT} \leftarrow \frac{1}{n_T} \sum_{i=1}^{n_T} x_{ijT}$ for $j > d_S$.
$x_{iT} \leftarrow x_{iT} - \bar{x}_T$, for $i \in [1, ..., n_T]$.
$\hat{\theta}_T \leftarrow (x_T^\top x_T)^{-1} x_T^\top y_T$.
$\hat{\theta}_S \leftarrow (x_S^\top x_S)^{-1} x_S^\top y_S$.
$\alpha_S \leftarrow (n_S - d_S)/\|y_S - x_S \hat{\theta}_S\|^2$.
$\alpha_T \leftarrow (n_T - d_T)/\|y_T - x_T \hat{\theta}_T\|^2$.
Compute $\hat{\theta}_\alpha$ with Equation 4.

---

can rewrite Equation 7 as:

$$\mathcal{L} = \frac{1}{2}\mathcal{R}_\alpha(\theta) + \frac{n_S}{2}\log(\sigma_S^2) + \frac{n_S + n_T}{2}\log(2\pi) + \frac{n_T}{2}\log(\sigma^2) \tag{8}$$

This means that, in this case, the data-pooling approach is a reasonable way to approximate the MLE of $\theta$ by minimizing a simpler criterion. It suggests that $\alpha_T = \frac{1}{\sigma^2}$ and $\alpha_S = \frac{1}{\sigma_S^2}$ might be an optimal choice for the hyperparameter $\alpha$.

### 4.3 The data-pooling algorithm

From a practical point of view, the data-pooling estimator cannot be computed directly since it depends on variables that in reality are unknown, namely $\sigma_S^2$ and $\sigma^2$. Nevertheless, in order to apply it in practice, these variables can be estimated separately as $\hat{\sigma}_S^2 = \|y_S - x_S\hat{\theta}_S\|^2/(n_S - d_S)$ and $\hat{\sigma}^2 = \|y_T - x_T\hat{\theta}_T\|^2/(n_T - d_T)$. Another practical impairment for the result above is that it relies on the assumption that the new features follow a normal distribution with zero mean ($\mathbb{E}[X''] = 0$). This is especially important for the data-pooling loss in Equation 3, where we fill up the lacking observations in the source dataset with zeros ($x_S\mathbb{I}^\top$). We approach this issue by shifting the observations of the new features in the target data by their estimated mean. It is important to notice that by using this trick, we are also changing the resulting estimate of the bias $\theta_0$. It means that, after computing the data-pooling estimator, to use it for predictions on new data, it is necessary to apply the same shift to that data. To sum up all these steps, we describe the algorithm to compute $\hat{\theta}_\alpha$ from the observations of source and target data in Algorithm 1.

In short, the data-pooling approach is a simple and computationally cheap way of approximating the maximum likelihood estimator of $\theta$ by combining historical data and new observations. In the following section, we want to study the benefit of using the source dataset by comparing our approach to the basic estimator.

## 5 Transfer gain of data-pooling

The transfer gain is a measure defined in (Obst et al., 2021) to assess whether a transfer learning model which uses both the target and the source datasets performs better than the basic estimator which uses only the target dataset. It is measured as the difference between the generalization error of the basic estimator and that of the transfer learning approach based on a new unseen data point from the target distribution.

Suppose we get a new row vector $x$ containing $d_T$ features. According to the definition of the target data, the corresponding label $Y$ is distributed as $Y = x\theta + w$, where $w \sim \mathcal{N}(0, \sigma^2)$ is independent from everything else. If we have an estimator $\hat{\theta}$ for $\theta$ then we can predict the new label as $x\hat{\theta}$. The generalization error is then given by $\mathbb{E}[(Y - x\hat{\theta})^2]$. In this work, we define the transfer gain in terms of the data-pooling estimator with weight $\alpha$ as:

$$\mathcal{G}(x) = \mathbb{E}[(Y - x\hat{\theta}_T)^2] - \mathbb{E}[(Y - x\hat{\theta}_\alpha)^2] \tag{9}$$

Intuitively, a positive transfer gain indicates that using the source dataset improves the predictions in new unseen data from the target distribution. If it is negative, then it means that either the source data is not helpful or that the transfer learning approach is sub-optimal.

**Proposition 5.1** *Since both $\hat{\theta}_T$ and $\hat{\theta}_\alpha$ are unbiased estimators of $\theta$, then the transfer gain can be rewritten in terms of the difference of their variances (proof in supplemental material):*

$$\mathcal{G}(x) = x(\text{Var}(\hat{\theta}_T) - \text{Var}(\hat{\theta}_\alpha))x^\top \tag{10}$$

*Furthermore, by replacing the variances of $\hat{\theta}_T$ and $\hat{\theta}_\alpha$ in equation equation 10, it becomes:*

$$\mathcal{G}(x) = x[\sigma^2 \Sigma_T^{-1} - M^{-1}(\sigma_S^2 \alpha_S^2 \mathbb{I}\Sigma_S \mathbb{I}^\top + \sigma^2 \alpha_T^2 \Sigma_T)M^{-1}]x^\top \tag{11}$$

The transfer gain formula in Equation 10 ties together the variances of $\hat{\theta}_\alpha$ and $\hat{\theta}_T$ in a way that if the variance of $\hat{\theta}_\alpha$ is smaller than that of $\hat{\theta}_T$, then the gain is strictly positive for any non-zero input vector $x$. It refers back to the problem of selecting $\alpha$ by adding a new objective: maximizing the transfer gain or at least making it positive. On top of that, Equation 11 expresses analytically this objective and can be used to study the signal of the gain.

We observe that by selecting $\alpha_S = \frac{1}{\sigma_S^2}$ and $\alpha_T = \frac{1}{\sigma^2}$ as mentioned in Section 4.2, the expression for the variance of $\hat{\theta}_\alpha$ simplifies greatly, becoming an updated version of $\text{Var}(\hat{\theta}_T)$. Based on this insight, we use the Woodbury's identity (Hager, 1989) to show that $\text{Var}(\hat{\theta}_T) - \text{Var}(\hat{\theta}_\alpha)$ is a positive semi-definite matrix (complete proof in the supplementary material). This results in the following theorem:

**Theorem 5.2** *Given any target dataset $(x_T, \boldsymbol{Y_T})$ and source dataset $(x_S, \boldsymbol{Y_S})$ as defined in Section 3, if we choose $\alpha_S = \frac{1}{\sigma_S^2}$ and $\alpha_T = \frac{1}{\sigma^2}$ then the transfer gain $\mathcal{G}(x)$ is non-negative for any given $x \in \mathbb{R}^{d_T}$. Therefore the generalization error of $\hat{\theta}_\alpha$ is upper-bounded by that of $\hat{\theta}_T$:*

$$\mathbb{E}[(Y - x\hat{\theta}_\alpha)^2] \leq \mathbb{E}[(Y - x\hat{\theta}_T)^2] \tag{12}$$

Theorem 5.2 leads to theoretical implications about the linear version of the incremental input learning problem and the data-pooling estimator. It says that by selecting the weights $\alpha_S = \frac{1}{\sigma_S^2}$ and $\alpha_T = \frac{1}{\sigma^2}$ the data-pooling estimator will be at least as good as the basic estimator. In other words, negative transfer is unlikely if we use data-pooling, so $\hat{\theta}_\alpha$ is a robust approach to combine the source and target datasets. Furthermore, this is true regardless of the amount of source and target samples, the number of extra features, or the value of their parameter $\theta''$.

It is even likely that using the source dataset via the data-pooling approach should give a positive gain. If we assume, for simplicity, the hypothetical case where the design matrices are orthogonal, so $\Sigma_S = I_{d_S}$ and $\Sigma_T = I_{d_T}$, then the variance of the data-pooling estimator becomes a diagonal matrix:

$$\text{Var}(\hat{\theta}_\alpha)_{ii} = \begin{cases} \frac{\sigma^2 \sigma_S^2}{\sigma^2 + \sigma_S^2}, & i \leq d_S \\ \sigma^2, & i > d_S \end{cases}$$

then it is straightforward to see that $\text{Var}(\hat{\theta}_\alpha)_{ii} \leq \text{Var}(\hat{\theta}_T)_{ii}$ and therefore $\mathcal{G}(x) \geq 0$. On top of that, if we assume that our new observation $x$ is such that $x_i \neq 0, i \leq d_S$, then the transfer gain is strictly positive. This means that if any of the features observable in the historical data are never zero, then the generalization error of $\hat{\theta}_\alpha$ is strictly lower than that of $\hat{\theta}_T$.

Theorem 5.2 could even generalize to a more complex setting where there is an arbitrary number of features that are irregularly observed. It can be done by defining multiple source datasets $(x_{S_i}, \boldsymbol{Y_{S_i}})$ where each one contains complete observations of a subset of these features, and extending the loss in Equation 3 by adding an extra term $\alpha_{S_i} \mathcal{R}_{S_i}$ for each dataset. Again, all the features with missing observations would have to be mean-shifted and the weights would be computed as $\alpha_{S_i} = \frac{1}{\sigma_{S_i}^2}$.

At this point, we know the theoretical upper bound of the error of the data-pooling estimator assuming that we have the values of $\sigma_S^2$ and $\sigma^2$. In the next section, we verify the result of Theorem 5.2 empirically and also try our transfer learning algorithm using real and simulated data.

# 6 Experimental Setup

We want to show how the data-pooling approach compares to other more complex SOTA transfer learning methods on multiple real-world datasets. In the following section, we explain in detail how we set up the experiments for this comparison. Additionally, in section 6.2 we present the setup of an ablation study using simulated data that we conduct to evaluate the impact of the theoretical assumptions about the data-pooling approach.

## 6.1 Real-life data experiment and SOTA comparison

In this experiment, we use multivariate real-life regression tasks to evaluate whether our theoretical results stand when the new inputs come from non-Gaussian distributions, and we compare the data-pooling method against other state-of-the-art heterogeneous transfer learning algorithms. The comparing methods are the Domain Specific Feature Transfer (DSFT) and its non-linear version (Wei et al., 2019), which are unsupervised and do not require any instance-correspondence, so they fit our incremental input learning setting as a pre-processing step. Both of them are based on learning a mapping from the historical features, which are present in both the source and the target datasets, to the new features. This mapping is used to fill up the values of the new feature in the source data, which is then combined with the target data to train a least-squares predictor. For the non-linear version of DSFT, a kernel is applied on the historical features before learning the mapping parameters. The hyperparameters are $\alpha = 10^5$ and $\beta = 1$ as recommended in the original paper and the kernel used is the radial basis function kernel (RBF), which has overall the best reported results according to the authors. As a baseline, we use the ordinary least-squares estimator computed using only the target dataset.

We use 9 multivariate regression datasets from the UCI repository (Dua & Graff, 2017), which are widely used in the literature for validating regression models. Each dataset contains a different amount of inputs in the source and target dataset and a wide variety of data distributions to showcase how the data-pooling estimator can work in more realistic settings. We separate each dataset into source, target, and test sets and remove a number of features from the source to emulate newly added features. The new features are selected by the largest Pearson correlation coefficient with respect to the regression label to ensure that the OLS baseline always uses the most relevant inputs for the task. We run the experiments with 3 and 5 features removed. All the details about the datasets are available in Table 1. The root-mean-squared error (RMSE) of each approach is computed using a separate test set. We follow two different data sampling strategies to evaluate our model using different amounts of source and target data samples, one for small data and another one for large data:

**Small data setup:** We fix the source and test datasets ($n_S$ and $n_{\text{test}}$ in Table 1) and sample multiple disjoint target datasets, one for each run, and use them to compute the transfer learning approaches and the OLS basic estimator. The number of runs is listed in Table 1.

**Large data setup:** For each dataset, we randomly sample 10% of the data for the test set, we fix $n_T$ at 100 or 300, and the remaining data goes to the source training set. This way, depending on the dataset, we can have very large source datasets (i.e. $\sim 40.000$ for *protein* dataset) or relatively larger target datasets (i.e. *concrete* and *energy*). We repeat the experiments 30 times, and each time we sample new source, target and test sets.

## 6.2 Ablation experiments

In this section, we define the setup to study the impact of the assumptions about our transfer learning approach in a practical setting and evaluate how it is influenced by the size of the target dataset $n_T$. We do so by computing the transfer gain empirically following the procedure explained in the next section. The subsequent sections describe the experiments where we analyze the effect of the choices for estimating the noise variance and shifting the new input by its sample mean.

Table 1: Dataset statistics.

| Dataset | $n_{\text{total}}$ | $n_S$ | $n_T$ | $n_{\text{test}}$ | $d_T$ | #runs |
|---------|--------|-----|-----|------|-----|-------|
| concrete | 1030 | 114 | 38 | 103 | 8 | 21 |
| energy | 768 | 114 | 38 | 76 | 8 | 15 |
| kin40k | 40000 | 114 | 38 | 4000 | 8 | 50 |
| parkinsons | 5875 | 150 | 50 | 587 | 20 | 50 |
| pol | 15000 | 168 | 56 | 1500 | 26 | 50 |
| protein | 45730 | 117 | 39 | 4573 | 9 | 50 |
| pumadyn32nm | 8192 | 186 | 62 | 819 | 32 | 50 |
| skillcraft | 3338 | 147 | 49 | 333 | 19 | 50 |
| sml | 4137 | 156 | 52 | 413 | 22 | 50 |

---

**Algorithm 2** Empirical Transfer Gain

---

**Input:** number of iterations $N$, test dataset $\mathcal{D}_{\text{test}}$
**for** $i = 1$ **to** $N$ **do**
    Randomly sample $\mathcal{D}_S$ and $\mathcal{D}_T$.
    Compute $\hat{\theta}_\alpha$ and $\hat{\theta}_T$.
    Compute the empirical transfer gain with Equation 13 (or Equation 14).
**end for**
Return the average over all $G_i$.

---

#### 6.2.1 Empirical transfer gain

Since the transfer gain depends on multiple sources of randomness, we have to estimate it empirically using multiple samples of source and target datasets. It is computed as the difference of the squared residuals of the prediction given by the basic estimator and the prediction given by the data-pooling estimator using a held-out test dataset $\mathcal{D}_{\text{test}}$ where all features are present. Each iteration $i$ corresponds to a different sample of training data that is used to compute $\hat{\theta}_\alpha$ and $\hat{\theta}_T$, which are then used to compute the difference of residuals as described in Equation 13. In the end, the empirical transfer gain is the result of the average over all the iterations. The complete procedure is detailed in Algorithm 2.

$$G_i = \frac{1}{n_{\text{test}}} \sum_{(x,y) \in \mathcal{D}_{\text{test}}} \left[ (y - x\hat{\theta}_T)^2 - (y - x\hat{\theta}_\alpha)^2 \right] \tag{13}$$

The way that $\hat{\theta}_\alpha$ is computed differs per experiment and is explained next.

#### 6.2.2 Estimating the noise variance

In the first simulation experiment, we want to test how accurately we can calculate $\hat{\theta}_\alpha$ if we estimate $\sigma^2$ and $\sigma_S^2$ (or $\alpha$) using the training data samples. For that, we simulate our data using the linear relationship $Y = 2 + 2X_1 - 2X_2 + w$, where $w, X_1, X_2 \sim \mathcal{N}(0,1)$, so the data fits our assumptions and all the parameters necessary to compute $\alpha$ and to accurately approximate the gain are known. In each iteration, we sample $n_S = 100$ source data points where $X_2$ is omitted. We vary $n_T$ from 5 to 30 samples to assess how the transfer gain changes with the size of the target data. We repeat the procedure in Algorithm 2 for each value of $n_T$. Finally, we use 1000 held-out samples for the test set and $N = 200$ iterations.

#### 6.2.3 Estimating the new input expectation

The goal of this experiment is to study the effect of shifting the target data by the sample mean when the assumption that it is zero-centered is not met. To achieve that, we compare the transfer gain of $\hat{\theta}_\alpha$ computed using the target data shifted by the real mean $\mathbb{E}[X'']$ versus $\hat{\theta}_\alpha$ computed using the target data shifted by

the sample mean $\bar{x}_T$ as described in Algorithm 1. Since we want to study this effect in isolation, we control for any other possible interfering factors by simulating the data as described in Experiment 1, except that the distribution of $X_2$ is changed to $\mathcal{N}(1,1)$. Again we follow the procedure described by Algorithm 2 to compute the transfer gain, only that at each iteration we also have to shift the test inputs by the mean computed from the target data, so Equation 13 becomes:

$$G_i = \frac{1}{n_{\text{test}}} \sum_{(x,y) \in \mathcal{D}_{\text{test}}} \left[ (y - x\hat{\theta}_T)^2 - (y - (x - \bar{x}_T)\hat{\theta}_\alpha)^2 \right] \tag{14}$$

In addition, we also look at how the variance of the sample mean affects the transfer gain by repeating the simulation with different target sample sizes $n_T$. Again, we use 1000 samples for the test set and $N = 200$ iterations.

## 7 Results

Here we analyze the results obtained from the experiments described in the previous section.

### 7.1 Real-life data experiment and SOTA comparison

The results of the experiments with multivariate datasets following the small and large data setups with 5 new features are shown in Table 7 and Table 2 respectively. Extra results with different numbers of new features and dataset sizes are shown in Appendix B.5. We use the Wilcoxon signed-rank test to determine when the difference between the models is statistically significant with a corrected p-value lower than 0.05. Underlined results signal whether data-pooling (DP) or non-linear DSFT has significantly lower error than the other. Similarly, italic is used to highlight the significance given by the test between DP and linear DSFT.

In the small data setup, our method outperforms the baseline in all cases except *concrete* and *energy*, where there was no significant difference between both. For 4 datasets, the data-pooling estimator has a significantly lower error than the linear DSFT and is outperformed in 1. In the remaining 4, there was no significant difference. In the comparison against the non-linear DSFT, data-pooling outperforms it in 2 datasets and is outperformed by it in 1 other. For the remaining 6 datasets, there is no statistically significant difference between both methods.

In the large data regime, DP outperforms OLS in 6 datasets. Only in the *concrete* dataset OLS outperforms DP, but also DSFT. Our approach also outperforms both variants of DSFT in the majority of the datasets. In the *energy* dataset, both versions of DSFT perform worse than the baseline, showing that the mapping used to impute the source dataset introduced substantial bias into the final predictor.

The results show that the transfer gain of our approach is mostly positive since it significantly outperforms the OLS estimator in most tasks, even though the new features follow arbitrary distributions and are not zero-centered. Only in 4 cases out of 36 cases (9 datasets × 4 experimental setups) it is outperformed by OLS. It shows that our theoretical upper bound on the generalization error can generalize to many real-life data distributions and that DP is robust against negative transfer learning. In addition, it shows that the data-pooling estimator outperforms the linear DSFT, and can perform on par with the more complex non-linear DSFT in the small data regime. In the large data regime, our approach outperforms both versions of DSFT largely, suggesting that imputing a very large source dataset is more difficult than computing the estimator directly through DP.

### 7.2 Estimating the noise variance

Figure 2a compares the empirical transfer gain when it is computed using the true values of $\sigma_S^2$ and $\sigma^2$ (the "real $\alpha$" line) and using their estimations from the data (the "estim. $\alpha$" line). We do so for different target dataset sizes shown in the x-axis. The result shows that the "real $\alpha$" curve does not go below zero, in accordance to Theorem 5.2, which means that the error of the data-pooling estimator computed on the test set is lower than that of the basic estimator. The transfer gain starts higher for small sizes of target data $n_T$

Table 2: RMSE of each approach following the **small data setup**. The 5 features with highest correlation with the label are removed from the source dataset. An asterisk (*) marks the datasets where there is no significant difference between DP and OLS.

| Dataset | ols | dsft | dsft-nl | dp |
|---|---|---|---|---|
| concrete* | $14.7 \pm 2.5$ | $14.5 \pm 2.68$ | $13.3 \pm 1.03$ | $14.3 \pm 2.81$ |
| energy* | $3.33 \pm 0.247$ | $3.91 \pm 0.434$ | $4.24 \pm 0.461$ | $3.32 \pm 0.257$ |
| kin40k | $1.12 \pm 0.062$ | $1.08 \pm 0.055$ | $1.06 \pm 0.0378$ | $1.07 \pm 0.0409$ |
| parkinsons | $16.8 \pm 5.32$ | $10.6 \pm 0.415$ | $10.8 \pm 0.414$ | $10.9 \pm 0.532$ |
| pol | $4.77\text{e}+09 \pm 3.38\text{e}+10$ | $120 \pm 108$ | $35.1 \pm 4.76$ | $36.1 \pm 5.86$ |
| protein | $0.825 \pm 0.11$ | $0.775 \pm 0.0975$ | $0.716 \pm 0.0233$ | $0.723 \pm 0.0321$ |
| pumadyn32nm | $1.49 \pm 0.142$ | $1.17 \pm 0.0788$ | $1.12 \pm 0.0446$ | $1.11 \pm 0.0393$ |
| skillcraft | $0.338 \pm 0.0439$ | $0.299 \pm 0.0215$ | $0.288 \pm 0.011$ | $0.29 \pm 0.0134$ |
| sml | $3.55 \pm 1.04$ | $3.1 \pm 0.949$ | $2.77 \pm 0.517$ | $2.81 \pm 0.521$ |

Table 3: RMSE of each approach following the **large data setup**. The 5 features with the highest correlation with the label are removed from the source dataset and the target dataset size is fixed at 100 samples. An asterisk (*) marks the datasets where there is no significant difference between DP and OLS and † marks when OLS is significantly better than DP.

| Dataset | ols | dsft | dsft-nl | dp |
|---|---|---|---|---|
| concrete[†] | $11 \pm 1$ | $11 \pm 0.986$ | $12.3 \pm 0.929$ | $11.1 \pm 0.93$ |
| energy* | $2.95 \pm 0.358$ | $3.31 \pm 0.423$ | $4.07 \pm 0.451$ | $2.94 \pm 0.336$ |
| kin40k | $1.04 \pm 0.0258$ | $1.02 \pm 0.022$ | $1.22 \pm 0.113$ | $1.02 \pm 0.0207$ |
| parkinsons | $11.1 \pm 1.16$ | $9.61 \pm 0.228$ | $10.1 \pm 0.293$ | $9.66 \pm 0.213$ |
| pol | $54.2 \pm 20.3$ | $49.5 \pm 46$ | $43.6 \pm 6.54$ | $31.8 \pm 0.763$ |
| protein | $0.729 \pm 0.0689$ | $0.728 \pm 0.0686$ | $0.68 \pm 0.0144$ | $0.679 \pm 0.0127$ |
| pumadyn32nm | $1.21 \pm 0.0732$ | $1.06 \pm 0.0672$ | $1.03 \pm 0.0359$ | $1.03 \pm 0.036$ |
| skillcraft* | $0.317 \pm 0.136$ | $0.658 \pm 1.3$ | $0.41 \pm 0.375$ | $0.409 \pm 0.373$ |
| sml | $2.87 \pm 0.919$ | $2.55 \pm 0.854$ | $2.34 \pm 0.229$ | $2.24 \pm 0.247$ |

and decreases fast as $n_T$ increases, to the point that the transfer gain gets closer to zero. We see that the gain is highest when $n_T$ is small, so the variance of $\hat{\theta}_T$ is larger and the use of the source dataset helps to reduce it for $\hat{\theta}_\alpha$. Finally, we see that the "estimated $\alpha$" curve overlaps almost perfectly with the real values, confirming our first hypothesis.

### 7.3 Estimating the new input expectation

Figure 2b shows the result of the simulation where the new feature is not zero-centered, so the data-pooling estimator has to be computed by shifting the observations of that feature by its estimated mean following Algorithm 1. For the comparison, we also plot the transfer gain for $\hat{\theta}_\alpha$ computed using the real mean. Again, we have the transfer gain in the y-axis and the target dataset size on the x-axis. We observe a difference between the gain curves of the true and the estimated mean, and the difference diminishes as $n_T$ increases. It represents the variance of the sample mean $\bar{x}_T$ that adds up to that of $\hat{\theta}_\alpha$ and makes the gain smaller. The variance of $\bar{x}_T$ is inversely proportional to $n_T$, so it also explains why the difference is larger for small $n_T$ and diminishes as $n_T$ grows. Nevertheless, the transfer gain is predominantly positive when $n_T$ is small. This shows that using the data-pooling approach can still be beneficial even in the case that the new features are not zero-centered.

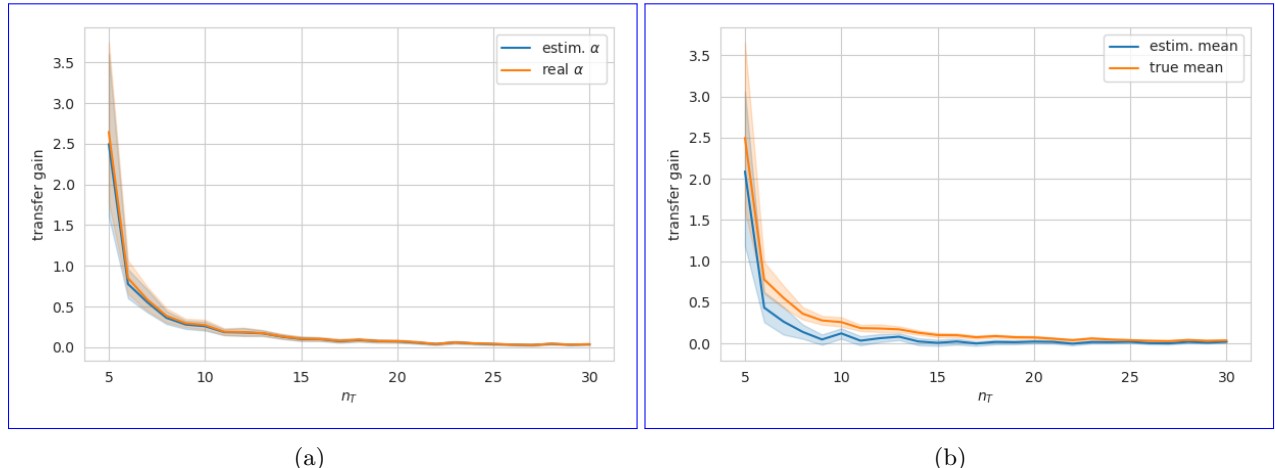

Figure 2: Comparison of the transfer gain of the data-pooling estimator when: (a) using the real value of $\alpha$ vs using the estimated value of $\alpha$; and (b) using the sample mean to shift the data vs using the true mean.

## 8 Conclusion

In this paper, we look at the problem of learning a predictive model after new input features are discovered, but there are only a few observations of them, while there are plenty of observations of historical data. We present a transfer learning approach to the linear regression version of this problem that is able to bridge the difference in the input dimensions of the source and the target datasets. We provide an in-depth theoretical study of our approach, proving an upper bound for its generalization error and its robustness against negative transfer learning. Through extensive empirical experiments using real-life datasets, we show that our approach performs consistently better than the baseline approach. The results hold for a wide variety of real-life distributions of new inputs, showing that the data-pooling approach still works when our theoretical assumptions are violated.

With respect to the state-of-the-art, our approach outperforms other linear transfer learning algorithms and performs on par with more complex non-linear ones, with the advantage of holding theoretical guarantees. It is also simple to implement and efficient, having the same computational complexity as the ordinary least-squares method. In addition, it does not have any hyperparameters to tune, so it is easy to apply to incremental input problems where the target data is limited.

Nevertheless, our theoretical bound on the generalization error of DP can still be improved by taking into account the estimations used for $\mathbb{E}[\boldsymbol{X''}]$, $\sigma_S$ and $\sigma$, which could help understand the effect of the dataset sizes $n_S$ and $n_T$ in the final predictor. Another future challenge is to study this problem in the non-linear case, such as classification with logistic regression or neural networks.

**Limitations:** Despite the positive results shown in the previous section, there are some clear limitations to the data-pooling approach and our theoretical results. The non-negative transfer gain proof relies on some assumptions which might not hold in practice:

1. independence between the new and the historical features ($\boldsymbol{X''}$ and $x_S$);

2. identical distribution of the new features and labels for both source and target data;

3. zero-centered Gaussian distribution of $\boldsymbol{X''}$;

4. linear model.

Assumption 1 can be violated, for example, if the data was generated by a non-additive model. We simulate this case in Appendix B.4 and indeed our approach leads to negative transfer more often than what we observe

in the results of Section 7.1, so important improvements can still be done in this direction. Assumption 2 means that we cannot guarantee non-negative transfer if there is a significant distribution shift between the source and the target datasets. Although data-pooling is resilient to distribution shifts in the historical features, it is sensitive to shifts in the distribution of the new features or if the parameter $\theta$ changes across the source and target data. We show empirically in Appendix B.3 that our approach can work with small shifts, but its transfer gain decreases as the shift grows. We find a workaround for Assumption 3 by centering the new features using the sample mean, but it introduces bias into DP which could lead to negative transfer when the target dataset is large enough (see Section 6.2.3). DP also relies on estimations of $\sigma_S^2$ and $\sigma^2$ might also introduce bias in the model. Finally, our proof is limited to linear models since it is based on a closed-form solution of the data-pooling loss. Therefore it is not trivial to translate it to non-linear cases such as neural networks and logistic regression.

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

## A   Proofs and derivations

### A.1   Proof of proposition 4.1

We want to find $\hat{\theta}_\alpha = \arg_\theta \min \mathcal{R}_\alpha(\theta)$, or $\hat{\theta}_\alpha$ such that $\nabla \mathcal{R}_\alpha(\hat{\theta}_\alpha) = 0$.

By applying simple calculus rules we have $\nabla \mathcal{R}_T(\theta) = -2x_T^\top(\boldsymbol{Y_T} - x_T\theta)$ and $\nabla \mathcal{R}_S(\theta) = -2\mathbb{I}x_S^\top(\boldsymbol{Y_S} - x_S\mathbb{I}^\top\theta)$, so we can write:

$$\nabla \mathcal{R}_\alpha(\theta) = -2\alpha_S \mathbb{I}x_S^\top(\boldsymbol{Y_S} - x_S\mathbb{I}^\top\theta) - 2\alpha_T x_T^\top(\boldsymbol{Y_T} - x_T\theta)$$

By equalling it to zero and solving for $\theta$ we obtain:

$$\hat{\theta}_\alpha = (\alpha_S \mathbb{I}x_S^\top x_S\mathbb{I}^\top + \alpha_T x_T^\top x_T)^{-1}(\alpha_S \mathbb{I}x_S^\top\boldsymbol{Y_S} + \alpha_T x_T^\top\boldsymbol{Y_T})$$

Let $\boldsymbol{H}_{\mathcal{R}_\alpha} = 2(\alpha_S \mathbb{I} x_S^\top x_S \mathbb{I}^\top + \alpha_T x_T^\top x_T)$ be the Hessian matrix of $\mathcal{R}_\alpha$. We assume that $x_T$ is obtained from observations of $d_T$ independent features, then the columns of $x_T$ must be linearly independent and thus, for any vector $v \in \mathbb{R}^{d_T}$ such that $v \neq \boldsymbol{0}$, $x_T v \neq \boldsymbol{0_{n_s}}$. This means that $v^\top x_T^\top x_T v > 0$ and so $x_T^\top x_T$ is positive definite. The same holds for $x_S^\top x_S$. Finally, $\alpha_S$ and $\alpha_T$ are both positive, so $\boldsymbol{H}_{\mathcal{R}_\alpha}$ is positive definite and therefore $\mathcal{R}_\alpha$ is strictly convex and $\hat{\theta}_\alpha$ is its global minimum. It also follows that the matrix $M = \alpha_S \mathbb{I} x_S^\top x_S \mathbb{I}^\top + \alpha_T x_T^\top x_T$ is positive definite since $M = \frac{1}{2}\boldsymbol{H}_{\mathcal{R}_\alpha}$, so the inverse $M^{-1}$ required to compute $\hat{\theta}_\alpha$ exists. $\square$

## A.2  Proof of proposition 4.2

Let $M = \alpha_S \mathbb{I} x_S^\top x_S \mathbb{I}^\top + \alpha_T x_T^\top x_T$ and knowing that $\theta' = \mathbb{I}^\top \theta$:

$$\begin{aligned}
\mathbb{E}[\hat{\theta}_\alpha] &= \mathbb{E}[M^{-1}(\alpha_S \mathbb{I} x_S^\top \boldsymbol{Y_S} + \alpha_T x_T^\top \boldsymbol{Y_T})] \\
&= \mathbb{E}[M^{-1}(\alpha_S \mathbb{I} x_S^\top \boldsymbol{Y_S}) + M^{-1}(\alpha_T x_T^\top \boldsymbol{Y_T})] \\
&= M^{-1}(\alpha_S \mathbb{I} x_S^\top \mathbb{E}[\boldsymbol{Y_S}]) + M^{-1}(\alpha_T x_T^\top \mathbb{E}[\boldsymbol{Y_T}]) \\
&= M^{-1}(\alpha_S \mathbb{I} x_S^\top x_S \theta') + M^{-1}(\alpha_T x_T^\top x_T \theta) \\
&= M^{-1}(\alpha_S \mathbb{I} x_S^\top x_S \theta' + \alpha_T x_T^\top x_T \theta) \\
&= M^{-1}(\alpha_S \mathbb{I} x_S^\top x_S \mathbb{I}^\top \theta + \alpha_T x_T^\top x_T \theta) \\
&= M^{-1}(\alpha_S \mathbb{I} x_S^\top x_S \mathbb{I}^\top + \alpha_T x_T^\top x_T)\theta \\
&= M^{-1} M \theta \\
&= \theta
\end{aligned}$$

$$\begin{aligned}
\mathrm{Var}(\hat{\theta}_\alpha) &= \mathrm{Var}(M^{-1}(\alpha_S \mathbb{I} x_S^\top \boldsymbol{Y_S} + \alpha_T x_T^\top \boldsymbol{Y_T})) \\
&= \mathrm{Var}(\alpha_S M^{-1} \mathbb{I} x_S^\top \boldsymbol{Y_S} + \alpha_T M^{-1} x_T^\top \boldsymbol{Y_T}) \\
&= \mathrm{Var}(\alpha_S M^{-1} \mathbb{I} x_S^\top \boldsymbol{Y_S}) + \mathrm{Var}(\alpha_T M^{-1} x_T^\top \boldsymbol{Y_T}) \\
&= \alpha_S^2 M^{-1} \mathbb{I} x_S^\top \mathrm{Var}(\boldsymbol{Y_S}) x_S \mathbb{I}^\top M^{-1} + \alpha_T^2 M^{-1} x_T^\top \mathrm{Var}(\boldsymbol{Y_T}) x_T M^{-1} \\
&= \alpha_S^2 (\sigma^2 + \|\theta''\|^2) M^{-1} \mathbb{I} x_S^\top x_S \mathbb{I}^\top M^{-1} + \alpha_T^2 \sigma^2 M^{-1} x_T^\top x_T M^{-1} \\
&= M^{-1}(\alpha_S^2(\sigma^2 + \|\theta''\|^2)\mathbb{I} x_S^\top x_S \mathbb{I}^\top + \alpha_T^2 \sigma^2 x_T^\top x_T) M^{-1}
\end{aligned}$$

## A.3  Proof of proposition 5.1

$$\begin{aligned}
\mathcal{G}(x) &= \mathbb{E}[(Y - x\hat{\theta}_T)^2] - \mathbb{E}[(Y - x\hat{\theta}_\alpha)^2] \\
&= \mathrm{Var}(Y - x\hat{\theta}_T) + \mathbb{E}[Y - x\hat{\theta}_T]^2 \\
&\quad - (\mathrm{Var}(Y - x\hat{\theta}_\alpha) + \mathbb{E}[Y - x\hat{\theta}_\alpha]^2) \\
&= \mathrm{Var}(Y - x\hat{\theta}_T) + (\mathbb{E}[Y] - x\mathbb{E}[\hat{\theta}_T])^2 \\
&\quad - (\mathrm{Var}(Y - x\hat{\theta}_\alpha) + (\mathbb{E}[Y] - x\mathbb{E}[\hat{\theta}_\alpha])^2) \\
&= \mathrm{Var}(Y - x\hat{\theta}_T) + (x\theta - x\theta)^2 \\
&\quad - (\mathrm{Var}(Y - x\hat{\theta}_\alpha) + (x\theta - x\theta)^2) \\
&= \mathrm{Var}(Y - x\hat{\theta}_T) - \mathrm{Var}(Y - x\hat{\theta}_\alpha) \\
&= \mathrm{Var}(Y) + \mathrm{Var}(-x\hat{\theta}_T) - (\mathrm{Var}(Y) + \mathrm{Var}(-x\hat{\theta}_\alpha)) \\
&= \mathrm{Var}(Y) - \mathrm{Var}(Y) + \mathrm{Var}(-x\hat{\theta}_T) - \mathrm{Var}(-x\hat{\theta}_\alpha) \\
&= \mathrm{Var}(-x\hat{\theta}_T) - \mathrm{Var}(-x\hat{\theta}_\alpha) \\
&= x(\mathrm{Var}(\hat{\theta}_T) - \mathrm{Var}(\hat{\theta}_\alpha))x^\top
\end{aligned}$$

## A.4 Proof of Theorem 5.2

Let $H = \text{Var}(\hat{\theta}_T) - \text{Var}(\hat{\theta}_\alpha)$ describe the difference between variances of $\hat{\theta}_T$ and $\hat{\theta}_\alpha$ such that $\mathcal{G}(x) = xHx^\top$. We want to show that $\mathcal{G}(x) \geq 0$ for any $x \in \mathbb{R}^{d_T}$, so it suffice to prove that $H$ is positive semi-definite. By selecting $\alpha_S = \frac{1}{\sigma_S^2}$ and $\alpha_T = \frac{1}{\sigma^2}$ we obtain $M = \frac{1}{\sigma_S^2}\mathbb{I}\Sigma_S\mathbb{I}^\top + \frac{1}{\sigma^2}\Sigma_T$ and:

$$H = \sigma^2\Sigma_T^{-1} - M^{-1}(\sigma_S^2\alpha_S^2\mathbb{I}\Sigma_S\mathbb{I}^\top + \sigma^2\alpha_T^2\Sigma_T)M^{-1}$$
$$= \sigma^2\Sigma_T^{-1} - M^{-1}\left(\frac{1}{\sigma_S^2}\mathbb{I}\Sigma_S\mathbb{I}^\top + \frac{1}{\sigma^2}\Sigma_T\right)M^{-1}$$
$$= \sigma^2\Sigma_T^{-1} - M^{-1}MM^{-1}$$
$$= \sigma^2\Sigma_T^{-1} - M^{-1}$$
$$= \sigma^2\Sigma_T^{-1} - \left(\frac{1}{\sigma_S^2}\mathbb{I}\Sigma_S\mathbb{I}^\top + \frac{1}{\sigma^2}\Sigma_T\right)^{-1}$$

Let $A = \frac{1}{\sigma^2}\Sigma_T$, $C = \frac{1}{\sigma_S^2}\Sigma_S$, $U = \mathbb{I}$ and $V = \mathbb{I}^\top$. $M$ can be seen as an updated version of $A$, so we can derive its inverse using the Woodbury identity (section 3 of (Hager, 1989)) which states that given $A$, $C$ and $A + UCV$ invertible matrices then $M^{-1} = (A + UCV)^{-1} = A^{-1} - A^{-1}U(C^{-1} + VA^{-1}U)^{-1}VA^{-1}$:

$$H = A^{-1} - (A + UCV)^{-1}$$
$$= A^{-1} - (A^{-1} - A^{-1}U(C^{-1} + VA^{-1}U)^{-1}VA^{-1})$$
$$= A^{-1}U(C^{-1} + VA^{-1}U)^{-1}VA^{-1}$$
$$= A^{-1}\mathbb{I}(C^{-1} + \mathbb{I}^\top A^{-1}\mathbb{I})^{-1}\mathbb{I}^\top A^{-1}$$

We know that $A$, $A^{-1}$, $C$, $C^{-1}$ are positive definite. We also know that $\mathbb{I}$ has full column rank and therefore $\mathbb{I}^\top A^{-1}\mathbb{I}$ is positive definite and so is $C^{-1} + \mathbb{I}^\top A^{-1}\mathbb{I}$ and its inverse. Finally, $\mathbb{I}(C^{-1} + \mathbb{I}^\top A^{-1}\mathbb{I})^{-1}\mathbb{I}^\top$ is positive semi-definite and thus, since $A^{-1}$ is symmetric, $H$ is positive semi-definite. $\square$

# B   Additional ablation studies

## B.1   Correlation experiment

An important difference between our approach and other state-of-the-art heterogeneous transfer learning algorithms, such as DSFT, is that we avoid learning a mapping from the old features to the new ones. So when there is little or no learnable relationship between the new and the old features our method should be able to perform better than DSFT. To investigate this hypothesis, we set up an experiment where we can control the relationship between the features so we can compare both methods in different settings.

In this experiment, we simulate data with a linear correlation between the input features to emulate a learnable relationship. The input variables are sampled from a joint Gaussian distribution with mean $\mu = (0, 1)^\top$ and covariance matrix $\Sigma$ such that $\Sigma_{ii} = 1$ for $i \in 1, 2$ and $\Sigma_{ij} = c$ for $i \neq j$ and $i, j \in 1, 2$. When $c = 0$, $X_1$ and $X_2$ are completely uncorrelated and when $c$ increases, so does the correlation between the two inputs, allowing us to control the strength of the correlation. The prediction label $Y$ is then computed as a linear combination of the inputs using the same parameters used in Section 6.2.3.

We fix the sizes of the source, target, and test datasets as $n_S = 100$, $n_T = 8$ and $n_{\text{test}} = 1000$, respectively. We vary $c$ between 0 and 0.9. For each value of $c$, we repeat the experiment 200 times resampling the source and target datasets while the test dataset remains fixed. The OLS computed on the target dataset is used as a baseline comparison.

Figure 3 shows the MSE computed on the test dataset with each approach for different values of the correlation coefficient $c$. As expected, the error for both linear and non-linear versions of DSFT is the highest when the input correlation is low, so it cannot learn an accurate mapping from $X_1$ to $X_2$. Their performance improves as $c$ increases, but is mostly larger than the OLS baseline, which means that the bias added by

using the inputted values of $X_2$ is still larger than the variance reduction expected from using the source data. Overall, this experiment shows that DSFT can only outperform DP on linear regression incremental input tasks when there is some significant (non-)linear relationship between the new and historical features.

We can also see in Figure 3 that the MSE of DP increases as $c$ increases. That happens because our approach expects the new inputs to be uncorrelated with the old inputs. This results in an increase of the bias of DP proportional to $c$, up to the point that it becomes larger than the variance reduction obtained from the source data when $c > 0.2$. At higher correlation values ($c > 0.7$), $X_1$ and $X_2$ are so similar that $Y$ can be predicted mainly by using $X_1$, so the source data becomes more and more useful. This explains the decrease in the MSE of the DP estimator at that point. Nevertheless, DP remains better than all other approaches for relatively small values of $c$, showing that there can be a positive trade-off between the variance reduction from the extra source data and the bias introduced by correlated inputs.

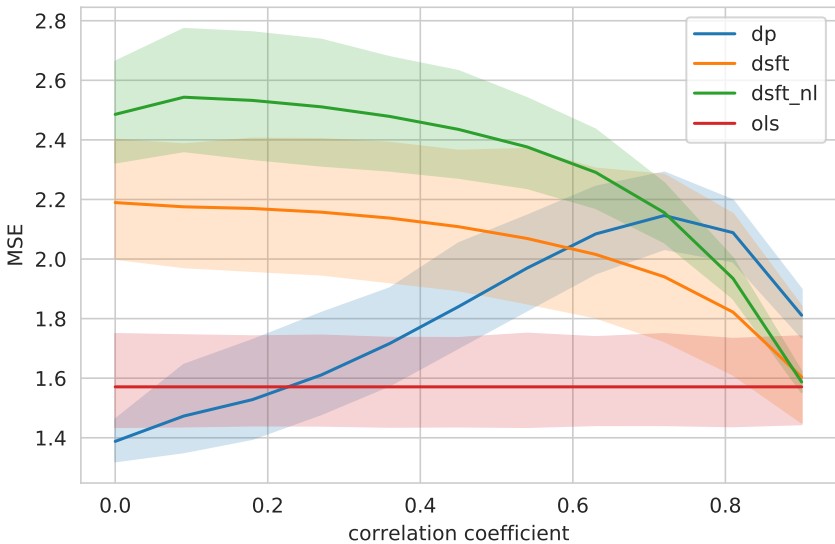

Figure 3: Correlation experiment: The MSE of both the data-pooling (*dp*) and the DSFT approaches for datasets generated with different correlation coefficients between the inputs.

### B.2 Detailed experiments with real-life data

For a detailed analysis, we selected three regression datasets from the UCI public repository: *Wine*, *Airfare* and *Concrete*; each one with different Pearson correlation coefficients measured between the input variables. *Wine* is the task of predicting the quality of red wines, *Concrete* is the task of predicting the resulting compressive strength measured in megapascal (MPa) of different concrete mixes and *Airfare* is the task of predicting the price of plane tickets based on the travel distance and the largest market share among all carriers. Table 4 gives further information about the variables chosen for the source and target datasets and Table 5 shows the Pearson correlation coefficients of each pair of variables. For each dataset, we fixed the source and the test datasets, and then sampled target datasets with different sizes ($n_T$), starting with 8 data points up to 30. For each value of $n_T$, we repeat the experiment multiple times, each time with a new independent target dataset. All the details about the settings selected for each dataset are stated in Table 5.

The results obtained using the *Wine*, *Concrete* and *Airfare* datasets are shown in figures 4, 5 and 6 respectively. On Figure 6, the non-linear DSFT gives the best results, suggesting that there is some exploitable non-linear relationship between the inputs *nsmiles* and *large_ms*. However, it does not confirm its superiority on the other two datasets (figures 4 and 5), where it displays similar performance to both DP and linear DSFT.

Table 4: Variables used for source and target data per dataset. $X_1$ appears in both source and target datasets, and $X_2$ only on the target dataset.

| Dataset | $X_1$ | $X_2$ | $Y$ |
|---------|-------|-------|-----|
| Wine | alcohol | volatile acidity | quality |
| Airfare | nsmiles | large_ms | fare |
| Concrete | Cement | Superplasticizer | Mpa |

Table 5: The Pearson correlation coefficients, the test and source dataset sizes and the number of repetitions for each dataset.

| Dataset | $\text{cor}(X_1, X_2)$ | $\text{cor}(X_1, Y)$ | $\text{cor}(X_2, Y)$ | $n_{\text{test}}$ | $n_S$ | #runs |
|---------|------------|------------|------------|-------|-------|-------|
| Wine | -0.20 | 0.48 | -0.39 | 700 | 100 | 25 |
| Airfare | -0.48 | 0.53 | -0.20 | 5000 | 100 | 50 |
| Concrete | 0.09 | 0.50 | 0.36 | 180 | 100 | 25 |

On the *Concrete* prediction task, DP outperforms both variants of DSFT in some moments, such as when the target dataset has 25 or more data points. The DP performance is comparable to that of the linear DSFT in all datasets, even *Wine* and *Airfare* where the correlation between inputs is higher.

These results show a pattern consistent with the simulations: the transfer gain is the largest when $n_T$ is small. In these cases, the size of the target dataset is too small and the variance of the basic estimator is the highest, so combining the source dataset using our transfer learning approach is more beneficial, resulting in a larger gain.

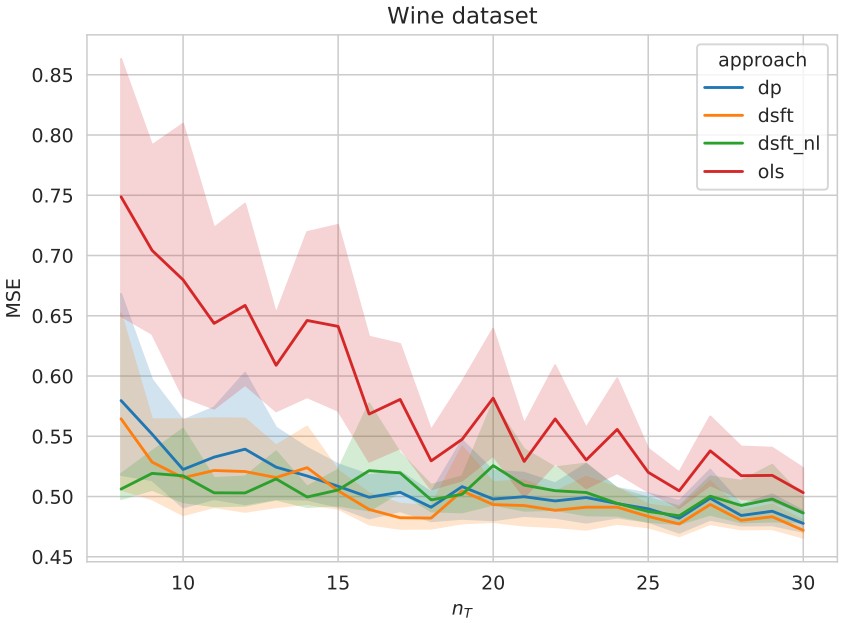

Figure 4: Comparison between the data-pooling (DP) and the DSFT approaches using the *Wine* dataset.

## B.3 Distribution shifts between source and target datasets

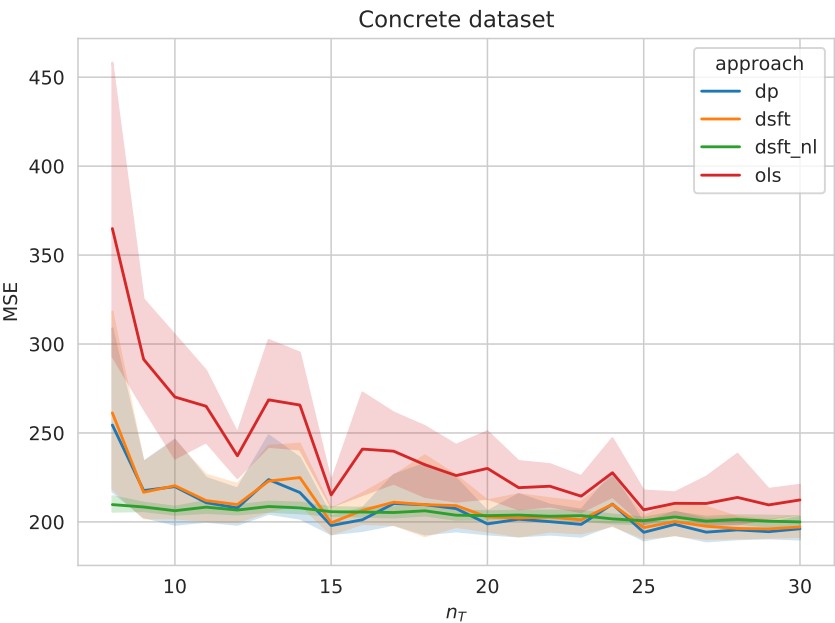

Figure 5: Comparison between the data-pooling (DP) and the DSFT approaches using the *Concrete* dataset.

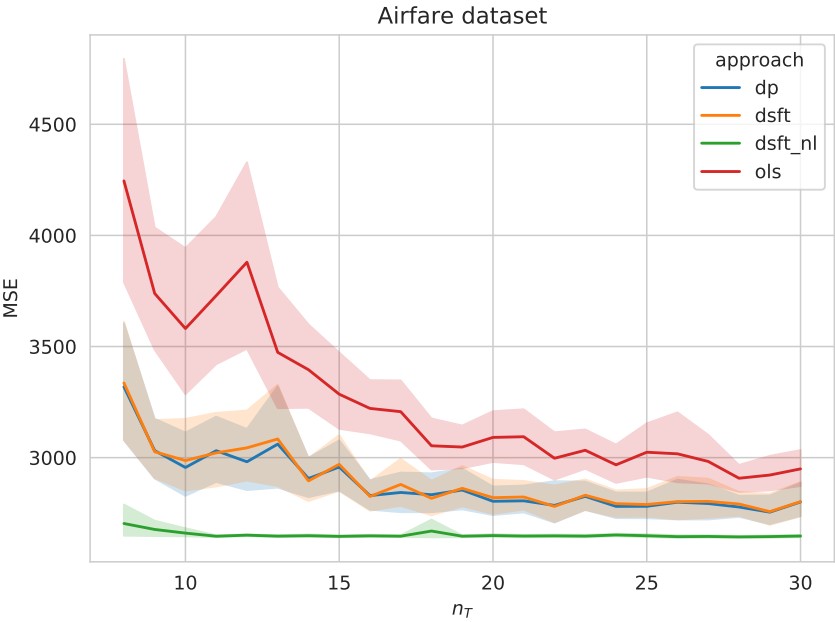

Figure 6: Comparison between the data-pooling (DP) and the DSFT approaches using the *Airfare* dataset.

An important assumption of our theoretical work is that the distribution of the new inputs remains the same for both the source and the target datasets, and we cannot guarantee the non-negative transfer gain bound if that does not hold. To demonstrate this problem, we simulate datasets following the same setup as Section 6.2.3, but we vary the mean of the new input in the target dataset $\mu_{1T}$ while it is fixed in the source dataset $\mu_{1S}$. We also fix the size of the source and target datasets at $n_S = 200$ and $n_T = 15$ to keep a similar proportion as the other experiments. The result in Figure 7 shows that the error of our estimator

increases quickly with the size of the shift, showing that the source dataset becomes less relevant for the linear regression task. Still, data-pooling copes with minor shifts in the distribution, and overall outperforms DSFT.

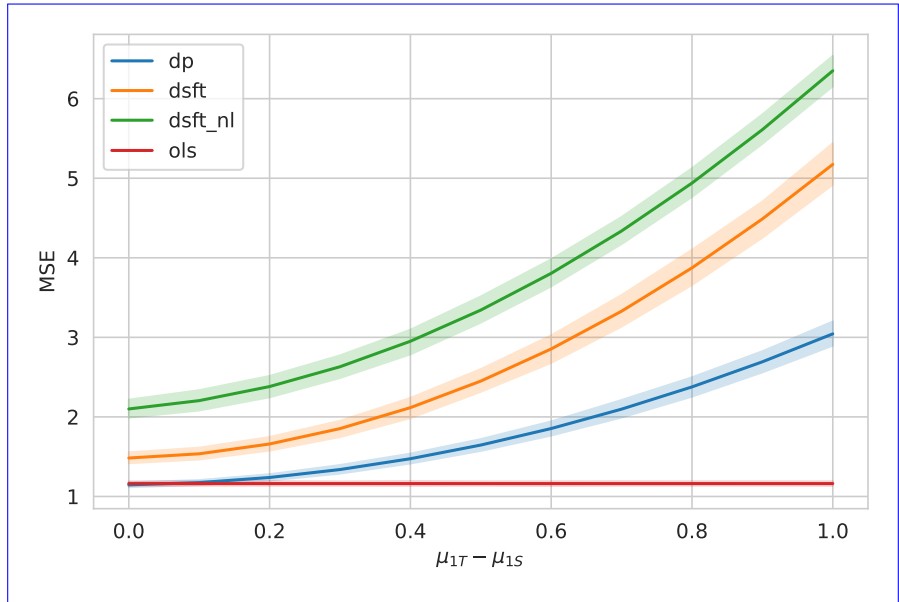

Figure 7: Result of simulation of increasing the difference of the mean of the new features between source and target datasets.

### B.4  Non-additive models

In our theoretical study, we assume that the new features are independent of the historical ones. One practical way this assumption could be violated is in the case of a non-additive model where the label depends on the product of a historical feature and a new feature. Since DP is a linear model, it is possible to compute this product and add it as a new feature, but then our independence assumption is broken. We can simulate this scenario by creating one such feature and adding it to the final label. We sort the features by their correlation with the label and we select the best one from the source set and from the target sets to create the product feature. For (non-linear) DSFT, we first compute the normal (additive) features for the source dataset and then use them to compute the product feature. We use the UCI benchmark datasets to keep the simulation as close as possible to reality. All the models are run following the *large data setup* described in Section 6.1.

The results presented in Table 6 show that DP performs consistently better than both versions of DSFT. In some cases such as *concrete*, *protein*, *skillcraft* and *sml*, DSFT's error is orders of magnitude higher than the OLS baseline, indicating that the values predicted by it to fill up the source dataset ended up biasing the final predictor heavily. As expected, the error of our approach increases w.r.t OLS's error, but still, it only gets outperformed in 3 out of 9 cases.

### B.5  Extra real-life data results

Here we present the extra results from Section 7.1. Table 7 shows the results following the small data setup with 3 new features, and Table 8 shows the results following the large data setup with $n_T = 300$. In Table 7, our approach performs consistently better than OLS and the linear DSFT with significantly lower error in 7 and 5 out of 9 datasets, respectively. The non-linear DSFT beats DP in 3 datasets, and there was no significant difference in the other 6.

Table 6: RMSE of each approach on the non-additive version of the UCI datasets. The target dataset size is fixed at 100 samples. An asterisk (*) marks the datasets where there is no significant difference between DP and OLS and † marks when OLS is significantly better than DP.

| Dataset | ols | dsft | dsft-nl | dp |
|---|---|---|---|---|
| concrete* | $11 \pm 0.996$ | $2.88\text{e}+03 \pm 1.09\text{e}+03$ | $4.9\text{e}+03 \pm 2.84\text{e}+03$ | $11 \pm 0.997$ |
| energy* | $2.87 \pm 0.388$ | $3.27 \pm 0.398$ | $4.04 \pm 0.457$ | $2.86 \pm 0.348$ |
| kin40k | $1.05 \pm 0.0266$ | $1.36 \pm 0.188$ | $1.58 \pm 0.149$ | $1.03 \pm 0.0246$ |
| parkinsons | $11.4 \pm 1.21$ | $10.5 \pm 0.837$ | $10.1 \pm 0.3$ | $9.72 \pm 0.258$ |
| pol† | $53.5 \pm 18$ | $133 \pm 84.4$ | $240 \pm 47.1$ | $180 \pm 46.7$ |
| protein† | $0.735 \pm 0.0679$ | $162 \pm 63$ | $52.6 \pm 8.7$ | $0.99 \pm 0.353$ |
| pumadyn32nm | $1.23 \pm 0.0793$ | $1.45 \pm 0.0583$ | $1.32 \pm 0.0626$ | $1.04 \pm 0.0406$ |
| skillcraft* | $0.344 \pm 0.245$ | $25.9 \pm 60.1$ | $9.18 \pm 2.57$ | $0.405 \pm 0.431$ |
| sml† | $3.05 \pm 0.965$ | $14.2 \pm 7.32$ | $31.5 \pm 14.3$ | $4.26 \pm 1.4$ |

Table 7: RMSE of each approach following the small data setup. The 3 features with highest correlation with the label are removed from the source dataset. An asterisk (*) marks the datasets where there is no significant difference between DP and OLS and † marks when OLS is significantly better than DP.

| Dataset | ols | dsft | dsft-nl | dp |
|---|---|---|---|---|
| concrete* | $14.7 \pm 2.5$ | $14.4 \pm 3.04$ | $13.1 \pm 0.7$ | $14.4 \pm 3.02$ |
| energy† | $3.33 \pm 0.247$ | $3.61 \pm 0.22$ | $3.59 \pm 0.22$ | $3.54 \pm 0.194$ |
| kin40k | $1.12 \pm 0.062$ | $1.06 \pm 0.0351$ | $1.06 \pm 0.036$ | $1.06 \pm 0.0294$ |
| parkinsons | $16.8 \pm 5.32$ | $10.7 \pm 0.357$ | $10.7 \pm 0.386$ | $10.7 \pm 0.441$ |
| pol | $4.77\text{e}+09 \pm 3.38\text{e}+10$ | $108 \pm 98.6$ | $36.3 \pm 6.26$ | $36.6 \pm 6.65$ |
| protein | $0.825 \pm 0.11$ | $0.767 \pm 0.117$ | $0.7 \pm 0.0186$ | $0.706 \pm 0.025$ |
| pumadyn32nm | $1.49 \pm 0.142$ | $1.13 \pm 0.0653$ | $1.09 \pm 0.0305$ | $1.09 \pm 0.0287$ |
| skillcraft | $0.338 \pm 0.0439$ | $0.27 \pm 0.0135$ | $0.257 \pm 0.0104$ | $0.26 \pm 0.0114$ |
| sml | $3.55 \pm 1.04$ | $2.98 \pm 0.852$ | $2.71 \pm 0.457$ | $2.78 \pm 0.493$ |

For the results with a larger target dataset in Table 8, we see that the OLS baseline benefits from the extra samples and so it outperforms DP in two cases, but DP is still superior in 6 others. In comparison with DSFT, DP was again better showing that it can make better use of the additional data.

Table 8: RMSE of each approach following the large data setup. The 5 features with the highest correlation with the label are removed from the source dataset and the target dataset size is fixed at 300 samples. An asterisk (*) marks the datasets where there is no significant difference between DP and OLS and † marks when OLS is significantly better than DP.

| Dataset | ols | dsft | dsft-nl | dp |
|---|---|---|---|---|
| concrete† | $10.5 \pm 0.782$ | $10.5 \pm 0.793$ | $11 \pm 0.554$ | $10.7 \pm 0.745$ |
| energy | $2.85 \pm 0.321$ | $2.92 \pm 0.3$ | $2.94 \pm 0.286$ | $2.84 \pm 0.318$ |
| kin40k | $1.01 \pm 0.0168$ | $1.01 \pm 0.015$ | $1.14 \pm 0.0631$ | $1.01 \pm 0.0146$ |
| parkinsons* | $9.59 \pm 0.248$ | $9.42 \pm 0.231$ | $9.79 \pm 0.267$ | $9.55 \pm 0.219$ |
| pol | $32.6 \pm 1.09$ | $31.2 \pm 0.594$ | $39.8 \pm 3.58$ | $31.1 \pm 0.419$ |
| protein | $0.683 \pm 0.0241$ | $0.681 \pm 0.0213$ | $0.668 \pm 0.00648$ | $0.668 \pm 0.00612$ |
| pumadyn32nm | $1.05 \pm 0.0384$ | $1.01 \pm 0.0321$ | $1.01 \pm 0.0301$ | $1.01 \pm 0.0309$ |
| skillcraft† | $0.354 \pm 0.351$ | $0.644 \pm 1.18$ | $0.399 \pm 0.358$ | $0.4 \pm 0.368$ |
| sml | $2.34 \pm 0.329$ | $2.22 \pm 0.277$ | $2.19 \pm 0.116$ | $2.14 \pm 0.11$ |

