# OpenReview forum: "Transfer Learning Across Datasets with Different Input Dimensions: An Algorithm and Analysis for the Linear Regression Case"
_TMLR — Rejected by TMLR_

### Review · Reviewer_gSHw · 2022-11-29

**Summary Of Contributions:**

In this paper, the authors study a variation of the heterogeneous transfer learning problem termed incremental input transfer learning. The main underlying idea of it is to assume that new input dimensions are added with time and that we would like to leverage on the historic data and on newly acquired data to learn a better prediction model. The latter’s performance is compared against the baseline given by the model learned from new data only. The authors make a set of simple assumptions for their model in linear regression setting and derive several insightful theoretical results from it. These theoretical results are further confirmed on both toy and UCI benchmarks.

**Audience:**

Yes

**Claims And Evidence:**

Yes

**Requested Changes:**

As expected, my requested changes are directly related to the weaknesses mentioned above.

1. I think for the simple model considered in this paper, it is crucial to identify the limitations of such approach. I believe that it may be as useful as the positive results presented in this paper as it seems to me that it may not work well in many real-world settings. I would encourage the authors to spend some time on designing pitfall scenarios for their approach to show its limits. Once again, I think that there is nothing wrong in simplicity if it allows to capture our intuition about the studied subject, but the results presented in Theorem 5.2 seems to lack the latter quality.

2. I wonder why the authors didn't consider other heterogeneous TL methods such as Yan et al. 2018 and the recent HTL baseline from Redko et al, NeurIPS'20. In general, this remark is also related to the following claim made by the authors: "but we argue that for this kind of approach to work it requires some exploitable relationship between historical and new features, whereas our approach works when there is no such relationship". My question here is: in what practical scenarios the newly acquired features will have to meaningful relationship to the historic features? What would be the value of such features then?

Some typos to fixed:

p.3 In this section, want -> we want
p.3 usupervised approach -> unsupervised
p.5 unobserved features X”, -> should be .


**Strengths And Weaknesses:**

**Strengths**

1. A theoretical study of incremental input transfer learning setting for the linear regression model

I think that it is important to lay down a theoretical foundation for a given computational model before trying it out on the real-world data and I appreciate the authors' effort made in this direction.

2. Empirical study highlighting the derived theoretical guarantees and their usefulness for UCI datasets

The evaluation includes one baseline proposed to tackle the same problem as well as one baseline giving the score for when no historic data is used. I believe that this is a bare minimum for any transfer learning paper (even though, some other candidates can be adapted to solve the same task as explained below)

**Weaknesses**

1. Considered model is rather simple, all main theoretical results are straightforward derivations from known linear regression results

Indeed, most of the theoretical results of this paper are clearly simple corollaries from linear regression analysis. This stems from the fact that the proposed model is essentially a sum of two least-squares terms for source and target data plus a multiplication of the source model parameters by a mask matrix zeroing out all unobserved entries for features. That being said, I don't mind simplicity of the derived results but I doubt that they generalize to more complicated settings and may lack to capture intuition behind such TL setting as explained below.

2. Some results seem counterintuitive wrt the existing theory of learning across different domains.

The cited work of Ben-David et al. considers a similar semi-supervised setting in the context of domain adaptation (Sections 5 and 6). Based on their theoretical analysis, they identified several regimes for the benefits that combined source and target error minimization can bring. They are captured in Fig. 1 and Theorem 3 of that paper and depend on the relative size of source and target domains. In this paper, the authors claim when discussing negative transfer in their setting after Theorem 5.2 :
 "Furthermore, this is true regardless of the amount of source and target samples, the number of extra features, or the value of their parameter $\theta''$."
Given this kind of statement, I wonder whether the model studied by the authors really can explain the success behind the learning phenomenon in such setting for real-world settings.

---

> ### Author Response · Authors · 2022-12-19
> **Small clarification**
>
> We thank the reviewer for the quick and insightful feedback.
> First of all, we will give a complete response to all the reviewers' comments once all of them have it submitted.
> In the meantime, we would like to ask for clarification on the second point of your requested changes, where you ask "in what practical scenarios the newly acquired features will have to meaningful relationship to the historic features?" Did you mean "[...] the newly acquired features will have *no* meaningful relationship [...]"?
>
> If your question is motivated by our statement "our approach works when there is no such relationship", we agree that it is inaccurate and we will soften it in the next version to "our approach works when this relationship is weak". In fact, in appendix B.1 we show empirically that our approach also works when there is a weak relationship, which can happen in common practical scenarios.
>
> To address your question, we propose an example: predicting income from features such as gender and education. If then we add work experience as a new feature, at least in theory it should not have any relationship with the previous features, but we would expect it to be a good predictor for income.
>
> Would that be enough to satisfy your request, or do you expect further empirical demonstrations?

---

> ### Author Response · Authors · 2023-02-08
> **Reply to the requested changes of Reviewer gSHw**
>
> We thank the reviewer again for his comments.
> For his first point, we have extended the conclusion section with a detailed discussion of the limitations of our theoretical results and DP approach.
> We also added extra experiments to showcase two possible pitfalls: non-additive features and input distribution shift. You can see them in Appendix B.4 and B.3.
>
> For your second question, we attempted to provide an answer to it in our previous reply. Please let us know whether it fulfilled your request.

---

### Review · Reviewer_N53g · 2022-12-14

**Summary Of Contributions:**

- The authors tackled an interesting problem, called incremental input transfer learning. More specifically, we have feature sets A and B. Source data only includes features set A but target data includes both feature sets A and B.
- The authors use joint optimization of source and target datasets using equation (2) and (3) to achieve the linear model that works with both source and target datasets. This method is very simple but intuitive.
- The authors provide some interesting experimental results even though the size of the data is too small.

**Audience:**

No

**Claims And Evidence:**

No

**Requested Changes:**

1. Classification vs Regression
- Is there a big difference to apply the proposed method to classification problems?
- In other words, do we need significant modifications to convert the proposed method to apply for the classification problem?
- If not, it would be good to show the impact of the proposed method in classification domains (and compare with the heterogeneous transfer learning baselines for the classification).

2. Figures
- It would be good if the authors can provide two figures.
(1) Explain the incremental input transfer learning (maybe in the introduction)
(2) Describe the proposed method
- Those two figures will significantly improve the readability of the paper.

3. Assumption of X''
- In Section 3, the authors said that source and target data follow the same conditional distribution between X and Y.
- Therefore, theta = [theta', theta''].
- In that case, how does the X'' follow the normal distribution, independently?
- I think X'' should be related to x_s and also follows the distributions of (d_T - d_S) portion of x_T.

4. Non-additive model
- Section 3 is only satisfied if the underlying model is an additive model.
- However, in reality, the label is not only dependent on the feature independently. They may be jointly dependent on the features.
- More specifically, if the label is dependent on the multiplication of feature x_i and x_j where x_j is only included in the target dataset, how can the proposed method deal with this case?

5. Table 1
- Why do the authors discard many samples instead of using them for source, target or test data?
- For instance, with kin40k, the sum of source and target data is less than 200 which is 0.5% of the entire data.
- It would be good to explain why the authors only use a very small amount of data for training sets (both source and target).
- Also, would it be good if the authors provide more extensive experiments with various ranges of source and target datasets?

6. Limitations
- This work has many limitations.
- It would be good to summarize the limitations of this work at the end of the paper. For instance,
- It is only applicable with an additive model,
- The authors only rely on the linear model,
- The results are only based on very small data,

7. Etc.
- It would be good to briefly explain about the "negative transfer learning" in the introduction.
- nit; it would be better to position figure 1 and 2 side by side (with smaller size).

**Strengths And Weaknesses:**

Strengths:
- The authors solve the interesting problem which can be practical in real-world setting.
- The proposed method is intuitive and working well in very small datasets regime.

Weakness:
- The proposed methods have too many limitations that it is hard to apply to real-world problems.
- More specifically, it is only applicable for the underlying distributions with additive models. Also, it is only verified with linear model and with very small datasets.
- The proposed method and theoretical supports are somewhat straightforward and too simple.
- Paper writing can be significantly improved.

---

> ### Author Response · Authors · 2022-12-19
> **Clarification about the requested changes**
>
> We thank the reviewer for the timely and thorough comments about our paper. We would like to ask for some clarifications about how we can accommodate your requests better. Hereby is our change plan for each request:
>
> >1. Classification vs Regression
> >
> >- Is there a big difference to apply the proposed method to classification problems?
> >- In other words, do we need significant modifications to convert the proposed method to apply for the classification problem?
> >- If not, it would be good to show the impact of the proposed method in classification domains (and compare with the heterogeneous transfer learning baselines for the classification).
>
> In order to apply our method to a classification problem, it would require significant modifications, since the least-squares method (which our algorithm is based on) is suboptimal for classification problems. A more adequate approach would involve using a logistic regression model, which is non-linear and does not have a closed-form solution like least-squares, so it would also require a new theoretical study on its own.
>
> Nevertheless, it is an interesting future work direction, and we can extend our conclusion by mentioning it.
>
>
> >2. Figures
> >
> >- It would be good if the authors can provide two figures. (1) Explain the incremental input transfer learning (maybe in the introduction) (2) Describe the proposed method
> >- Those two figures will significantly improve the readability of the paper.
>
> Thank you for the suggestion, we agree that it will improve the readability and we will include it in the next version of the paper.
>
>
> >3. Assumption of X''
> >
> >- In Section 3, the authors said that source and target data follow the same conditional distribution between X and Y.
> >- Therefore, theta = [theta', theta''].
> >- In that case, how does the X'' follow the normal distribution, independently?
> >- I think X'' should be related to x_s and also follows the distributions of (d_T - d_S) portion of x_T.
>
>
> We do assume in Section 3 that X'' is independent from x_S (the new features are independent from the historical ones) and we agree to clearly state that assumption in Section 3, although we demonstrate empirically in our results that our method also works in cases where this assumption is violated.
>
> About the (d_T - d_s) portion of x_T, you are right that it should come from the same distribution as X'', that is a common assumption for regression models (iid samples). We propose to clarify that in Section 3.
>
>
> >4. Non-additive model
> >
> >- Section 3 is only satisfied if the underlying model is an additive model.
> >- However, in reality, the label is not only dependent on the feature independently. They may be jointly dependent on the features.
> >- More specifically, if the label is dependent on the multiplication of feature x_i and x_j where x_j is only included in the target dataset, how can the proposed method deal with this case?
>
> A simple way to deal with this situation would be to include x_i * x_j as another new feature in the target dataset.
> That would violate the independence assumption between x_S and X'', but we believe it would not always prevent our approach from working.
> We propose to include in our ablation study a simulation experiment to show how our approach would behave in comparison to the other non-linear baseline.
>
>
> >5. Table 1
> >
> >- Why do the authors discard many samples instead of using them for source, target or test data?
> >- For instance, with kin40k, the sum of source and target data is less than 200 which is 0.5% of the entire data.
> >- It would be good to explain why the authors only use a very small amount of data for training sets (both source and target).
> >- Also, would it be good if the authors provide more extensive experiments with various ranges of source and target datasets?
>
> We use small target datasets since that is the common scenario where transfer learning would be necessary.
> Nevertheless, we propose to extend our experiments in Section 6 with extra experiments using larger source and target datasets (n_s \in {500, 1000, 5000, 10.000} and n_t \in {100, 200}) using *kin40k*, *pol* and *protein*.
>
>
>
> >6. Limitations
> >
> >- This work has many limitations.
> >- It would be good to summarize the limitations of this work at the end of the paper. For instance,
> >- It is only applicable with an additive model,
> >- The authors only rely on the linear model,
> >- The results are only based on very small data,
>
> We will extend the conclusion section with a list of the assumptions of our theoretical work (i.e. linear model, gaussian data, independence between new and historical features). The limitations related to the additive model and dataset size, we intend to address through the experiments mentioned in the answers to your requests 4 and 5 respectively.
>
>
> Would all the aforementioned changes suffice to meet your criteria to accept the paper?

---

> > ### Comment · Reviewer_N53g · 2023-01-09
> > **Response to the authors**
> >
> > Thanks for your response.
> >
> > Unfortunately, I cannot say that the revised manuscript will meet my acceptance criteria.
> > I will decide after checking the revised manuscript once the authors submit.
> >
> > It would be great if the authors can address my point 3 (Assumptions), 4 (non-additive models), and 6 (Limitations) well to make the proposed method more general and practical.
> > Also, for the point 5 (small data), it would be good if we can check the generalizability of the proposed method with the entire data in Table 1 (not subsampling).
> >
> > Thanks again.

---

> > > ### Author Response · Authors · 2023-02-08
> > > **Reply to the requested changes of Reviewer N53g**
> > >
> > > In the new revision of the paper, we addressed points 1, 2, and 6 as we mentioned in the previous reply.
> > > For point 5 we re-ran the experiments using the entire datasets (no subsampling) in a cross-validation manner with 30 repetitions. We did so with the target dataset fixed at 100 and 300 samples.
> > >
> > > For point 4 we also added an experiment in Appendix B.3 simulating a distribution shift for the new features, showing that our model is sensitive to it, but can cope with small shifts.
> > >
> > > For point 3 we added an experiment simulating non-additive features using the UCI datasets and our model was able to cope with it in 6 out of 9 datasets. The results are in Appendix B.4.

---

### Review · Reviewer_sNW3 · 2023-01-28

**Summary Of Contributions:**

In this paper, authors study the problem of learning a predictive model after new input features are discovered, but there are only a few observations of them, while there are plenty of observations of historical data. They propose the data-pooling method that learns an estimator that jointly minimizes the squared error on the source and target data. To tackle the problem of difference in dimensions author restrict the estimator to relevant covariates on source data. Results on real-life datasets show that DP performs consistently better than the baseline approach (OLS and DSFT).

**Audience:**

Yes

**Claims And Evidence:**

Yes

**Requested Changes:**

It would be great if authors can address weaknesses from above.

**Strengths And Weaknesses:**

**Strengths**

- Simple framework to study the problem of transfer learning where the input dimensions differ.
- Author present theoretical study showing that DP Is robust to negative transfer when the number of samples very small.

**Weaknesses**
- A detailed description of the DSFT method should be included in the paper. The discussion of this method is restricted to only empirical comparison. It would be interesting to see what are the conceptual differences in the methodology.
- Can the proposed method be extended to classification? Are there some direct implications?
- Also author assumes that the distribution of covariates and labels across the overlapping dimensions remains invariant. It would be great to include some experiments/discussion on settings where such assumptions may be violated.

---

> ### Author Response · Authors · 2023-02-08
> **Reply to the requested changes of Reviewer sNW3**
>
> Thank you for the insightful comments. Here is how we attempt to accommodate each of your requested changes:
>
> > A detailed description of the DSFT method should be included in the paper. The discussion of this method is restricted to only empirical comparison. It would be interesting to see what are the conceptual differences in the methodology.
>
> We extended the literature review section with a more detailed description of the DSFT method and we compare it methodologically with our method.
>
> > Can the proposed method be extended to classification? Are there some direct implications?
>
> As we mentioned in the first reply to reviewer N53g, it would require significant modifications to apply our method to classification.
> We include a mention of this option as future work in the conclusion section.
>
> >Also author assumes that the distribution of covariates and labels across the overlapping dimensions remains invariant. It would be great to include some experiments/discussion on settings where such assumptions may be violated.
>
> The reviewer is correct in that observation. We extended the conclusion with a part about the limitations of our method where we discuss how that assumption can impact our approach in practice. We also added an experiment in Appendix B.3 simulating a distribution shift in the new features across source and target domains.

---

### Author Response · Authors · 2023-02-08
**Comment on the new version of the paper**

Dear reviewers,

Thank you very much for reading our paper and for the detailed feedback. We have done a major update to the paper in an attempt to fulfill all the requested changes. Hereby the list of the main changes for this revision:
- We extended the conclusion section with a paragraph about the limitations of our work where we list the 4 major assumptions of our theoretical work and discuss how each one of them could impact our approach in practice.
- We added extra experiments to demonstrate empirically the possible pitfalls of our method:
  - a simulation of a non-additive model following the suggestion of reviewer N53g and using the UCI datasets (Appendix B.4)
  - a simulation of a distribution shift between $X''$ and the observations in the $(d_T-d_S)$ part of $x_T$ (Appendix B.3)
- We added extra experiments using all the data from each of the 9 UCI datasets (Sections 6.1 and 7.1 and Appendix B.5). So the source and target datasets are considerably larger than in our original experiments.
- We extended the literature review section with a more detailed description of the DSFT method.

Other minor changes:
- Table 1: corrected columns $n_{\text{test}}$ and "*#runs*" for concrete and energy datasets
- Replace MSE for RMSE in the result tables for better readability (fewer rows with exponential notation)
- While implementing the new experiments we also made some improvements in the code which slightly changed the original results, but the main conclusions remain the same. This version shows the latest results. We will publish the code for all experiments once the review is over.
- Added image illustrating the incremental input problem in the introduction
- Fixed typos and other minor formatting issues.

We hope that these changes can accommodate all your requests.

---

### Decision · Action_Editors · 2023-04-03

**Recommendation:** Reject

**Comment:**

The manuscript introduces a transfer learning algorithm that integrates novel and past data with different input dimensions. The aim is to develop a predictive model using limited observations of newly discovered input features, while a wealth of historical data is available. The manuscript focuses on the linear regression problem, and the classification problem would be a future work. The proposed data-pooling method learns an estimator that jointly minimizes the squared error on the source and target data. To address the problem of input dimensionality mismatch, the method restricts the estimator to relevant covariates on source data. The manuscript evaluates the proposed method against two baseline approaches, Ordinary Least Squares and Domain Specific Feature Transfer (Wei et al., 2019), on nine multivariate regression datasets from the UCI repository.

The reviewers acknowledged that the manuscript addresses an important problem but expressed concern about the feasibility of the assumptions made. While they agreed that the manuscript's argument holds up under these assumptions, they felt that the assumptions were too restrictive to be applicable to real-world scenarios. As a result, they were uncertain if the readers of TMLR would find the manuscript relevant. The two primary assumptions were: A) that all newly observed features are completely independent of the original features, and B) that the relationship between feature labels in the real world can be expressed as an additive model.

For point A) the authors have added an experiment simulating non-additive features using the UCI datasets and the proposed approach was able to cope with it in 6 out of 9 datasets. The results are in Appendix B.4.

For point B) the authors have also added an experiment in Appendix B.3 simulating a distribution shift for the new features, showing that the proposed model is sensitive to it, but can cope with small shifts.


**Audience:**

I would not be able to answer positively to this acceptance criteria as reviewers agreed that they do not see any strong insights into the studied problem given the simplicity of its formulation and the assumptions were too restrictive to be applicable to real-world scenarios

**Claims And Evidence:**

The reviewers acknowledged that the manuscript addresses an important problem of developing a predictive model using limited observations of newly discovered input features, while a wealth of historical data is available. They agreed that the manuscript's argument holds up under the assumptions but expressed concern about the feasibility of the assumptions made.

---

> ### Author Response · Authors · 2023-04-17
> **Comment on Action Editors Decision**
>
> Dear action editor,
>
> Thanks for the efforts put into reviewing our paper and making a final decision. We are however slightly disappointed with the feedback we received. We have put a lot of effort into addressing all the changes requested by the reviewers, such as clarifying the limitations of our theoretical study, so that no unrealistic claims are made, and including extra experiments to show where our algorithm does and does not work. In your comment, you do acknowledge that our extra experiments cover the reviewers’ concerns and address the issue of the limitations, but no comments are given on whether those new experiments are not satisfactory, and if so, what is still lacking. Additionally, two of the reviewers are positive about the audience criteria, therefore some individuals within the journal audience are interested in our findings. Of course, this email is not intended to change your decision, but more to understand why all the extra effort we put in was not sufficient.
>
> Thank you in advance for your response,
>
> The authors